# GLUT4 Trafficking and Storage Vesicles: Molecular Architecture, Regulatory Networks, and Their Disruption in Insulin Resistance

**DOI:** 10.3390/ijms26157568

**Published:** 2025-08-05

**Authors:** Hana Drobiova, Ghadeer Alhamar, Rasheed Ahmad, Fahd Al-Mulla, Ashraf Al Madhoun

**Affiliations:** 1Department of Pathology, College of Medicine, Kuwait University, Jabriya 24923, Kuwait; hana.drobiova@ku.edu.kw; 2Immunology and Microbiology Department, Dasman Diabetes Institute, Dasman 15462, Kuwait; ghadeer.alhamar@dasmaninstitute.org (G.A.); rasheed.ahmad@dasmaninstitute.org (R.A.); 3Translational Research Department, Dasman Diabetes Institute, Dasman 15462, Kuwait; 4Animal and Imaging Core Facilities, Dasman Diabetes Institute, Dasman 15462, Kuwait

**Keywords:** GLUT4 storage vesicles, GSVs, insulin-responsive vesicles, IRVs, GLUT4, type 2 diabetes, intracellular trafficking, exocytosis and endocytosis

## Abstract

Insulin-regulated glucose uptake is a central mechanism in maintaining systemic glucose homeostasis, primarily occurring in skeletal muscle and adipose tissue. This process relies on the insulin-stimulated translocation of the glucose transporter, GLUT4, from specialized intracellular compartments, known as GLUT4 storage vesicles (GSVs), to the plasma membrane. Disruption of this pathway is a hallmark of insulin resistance and a key contributor to the pathogenesis of type 2 diabetes. Recent advances have provided critical insights into both the insulin signalling cascades and the complex biogenesis, as well as the trafficking and fusion dynamics of GSVs. This review synthesizes the current understanding of the molecular mechanisms governing GSV mobilization and membrane fusion, highlighting key regulatory nodes that may become dysfunctional in metabolic disease. By elucidating these pathways, we propose new therapeutic avenues targeting GSV trafficking to improve insulin sensitivity and combat type 2 diabetes.

## 1. Introduction

The prevalence of prediabetes and diabetes is increasing worldwide, with around 1 in 9 (589 million) adults (age 20–79 years) living with diabetes, and the number is expected to rise to 853 million by 2050 [1]. Prediabetes stands for the transitional phase between normal blood glucose levels and the diabetic phenotype and occurs because of impaired glucose homeostasis, called peripheral insulin resistance. Insulin resistance, characterized by the reduced responsiveness of peripheral tissues, particularly skeletal muscles and adipose tissues, to insulin, is a central feature of metabolic syndrome pathophysiology and is an early event in the development of cardiovascular disease and type 2 diabetes (T2D) in humans [2,3]. It is also associated with other diseases, including obesity, fatty liver steatosis, polycystic ovarian syndrome, hypertension, atherosclerosis, some forms of cancer, neurodegenerative diseases, and frailty [4,5,6,7]. Efficient regulation of glucose homeostasis requires the coordinated function of insulin-secreting pancreatic β-cells and insulin-sensitive peripheral tissues. Insulin, along with other hormones, plays a crucial role in maintaining whole-body glucose homeostasis by reducing hepatic glucose production and promoting glucose uptake in muscle and adipose tissues. This is achieved by the activation of intracellular signalling pathways that promote the trafficking of the facilitative glucose transporter (GLUT4; also known as solute carrier family 2 member 4) (Figure 1). GLUT4 is stored in specialized compartments, known as GLUT4 storage vesicles (GSVs), which translocate to the plasma membrane in adipocytes, or sarcolemma, and the traverse tubule membrane in myocytes, in response to insulin stimulation [8]. The translocation of GLUT4 to the plasma membrane increases glucose uptake by 10- to 30-fold [9,10,11]. GLUT4-mediated glucose uptake constitutes the rate-limiting step in insulin-stimulated glucose disposal, which has been confirmed by several independent pieces of evidence, including transgenic and knockout mice models [12] and in vivo human nuclear magnetic resonance investigations [13]. Therefore, insulin resistance may represent the primary defect in the development of T2D. Insulin resistance in adipose and skeletal muscle cells may result from a defect in insulin signalling, abnormal GLUT4 trafficking, or a combination of both. Therefore, elucidating the mechanistic processes of GLUT4 regulation is essential for understanding the molecular basis of insulin resistance and diabetes, as well as for developing novel potential therapeutic approaches. Hence, this review will focus on the composition, biogenesis, and insulin signalling that facilitate GSV mobilization and trafficking, examining their regulation under healthy physiological conditions and the alterations that occur in the context of insulin resistance.

## 2. Biogenesis and Molecular Composition of GSVs

Under basal conditions, only around 5–10% of the GLUT4 protein is present in the plasma membrane of adipose and muscle cells, while the vast majority, >90%, is retained intracellularly within “GLUT4 storage vesicles (GSVs)”, also referred to as “insulin-responsive vesicles (IRVs)” [14,15]. In response to insulin, these vesicles translocate to the plasma membrane, where they facilitate glucose uptake. Mathematical modelling of GLUT4 trafficking predicts two major sources of GLUT4 translocation to the plasma membrane, namely the endosomes and specific intracellular insulin-responsive compartment [16]. The model suggests that this specialized compartment is the major source of GLUT4 exocytosis in response to insulin stimulation [17,18]. This is supported by the observation that GLUT4 release from this compartment is undetectable in the absence of insulin, emphasizing the essential role of insulin in GLUT4 exocytosis [19,20,21,22]. In contrast to this static model, the dynamic equilibrium model proposes continuous GLUT4 cycling between the intracellular compartments and the plasma membrane, even in the absence of insulin, with a tendency toward intracellular sequestration [23,24]. However, the differences between these two models can be reconciled by evidence showing that insulin promotes GLUT4 exocytosis from both compartments [21,25,26]. The existence of these dual mechanisms is supported by live-cell imaging studies, biochemical and electron microscopy data analysis of cultured 3T3-L1 adipocytes and mouse muscle tissues [11,15,27,28,29,30,31].

Biochemical and immunoelectron microscopy studies have demonstrated that up to 75% of intracellular GLUT4 is sequestered within small vesicles (50–80 nm in diameter, approximately 80S) and short tubules [32,33], with the remainder being associated with large, rapidly sedimenting intracellular membranes, probably corresponding to endosomes and trans-Golgi network (TGN) compartments. These small GLUT4-containing vesicles can be biochemically isolated from endosomes and other organelles [30,34,35]. Purification, based on their physicochemical properties, revealed that these vesicles do not constitute a homogeneous population, but consist of two distinct subtypes: (1) IRVs, which in response to insulin, translocate to the plasma membrane, with nearly 100% efficiency, and serve as a pre-existing reservoir in unstimulated cells [36], and (2) a pool of ubiquitous intracellular transport vesicles that are not mobilized to the plasma membrane by insulin. In adipocytes, these insulin-unresponsive vesicles are marked by cellugyrin, a four-transmembrane protein, with currently unknown physiological roles [37]. This is vital as it suggests that the primary function of insulin may not be the initiation of vesicle formation or budding, but rather the mobilization of an already existing pool of GSVs. Furthermore, this mobilization cannot be explained by insulin-mediated membrane fusion alone; instead, a more appropriate mechanism involves insulin-stimulated cleavage of the tether containing a UBX domain for the GLUT4 (TUG) protein, which plays a central role in the release of GSVs from their anchored state [38]. This mechanism will be further discussed below.

It is hypothesized that a comprehensive analysis of the IRV protein or lipid composition may shed light on the mechanisms underlying their insulin sensitivity. In agreement with this idea, techniques such as immunoadsorption and other vesicle fractionation methods have facilitated the identification of key protein components within GLUT4-containing vesicles (Table 1). Despite the extensive biochemical enrichment, which can be assessed using these methods, the heterogeneity within GSVs limits the identification of all vesicle-associated proteins. Moreover, immunoadsorption techniques often strip away peripheral membrane proteins, impeding the discovery of downstream insulin signalling intermediates [39].

A systematic proteomic analysis of highly purified IRVs, specifically excluding cellugyrin-containing, insulin-unresponsive compartments, was conducted by Jedrychowski et al. [35]. This study identified three principal cargo proteins, GLUT4, insulin-responsive aminopeptidase (IRAP), and sortilin, as being abundantly present in the IRVs and that exhibited insulin responsiveness, consistent with previous studies of small GLUT4 vesicle preparations [32,34,36,38,39,40,41,42]. Additionally, Jedrychowski et al. identified that low-density lipoprotein receptor-related protein 1 (LRP1) is also an IRV component [35]. Although LRP1 is known to be translocated to the plasma membrane in response to insulin in adipose cells, it was overlooked in previous proteomic screenings, likely due to its substantial size (4544 amino acids) [39].

Although cellugyrin-positive transport vesicles share a similar protein composition to IRVs [35], they contain a markedly lower amount of key proteins [15]. This disparity suggests that GLUT4, IRAP, and possibly other core IRV proteins are actively and selectively sorted into IRVs. The underlying potential mechanisms of this selective sorting will be discussed in subsequent sections. Among IRV components, IRAP is possibly the most abundant component, and constantly colocalizes with GLUT4 throughout its intracellular journey [15,43,44]. While the physiological and functional role of IRAP in adipose and muscle tissues is not clearly understood, its presence in IRVs may enhance insulin sensitivity through its interaction with the AKT substrate of 160 kDa (AS160, also known as TBC1D4), a Rab GTPase-activating protein [34,45,46], and/or through its involvement in protein sorting via its recruitment from endosomes into GSVs [47].

Sortilin, a mammalian homolog of the yeast vacuolar sorting receptor, Vps10p, is a key IRV component. As a multifunctional sorting and signalling receptor, sortilin plays a key role in recruiting cargo proteins, such as GLUT4 and IRAP, into GSVs during vesicle budding, and interacts directly with LRP1 [38,39]. In adipocytes and muscle cells, sortilin is highly enriched in IRVs and may drive compartment formation through luminal domain-mediated protein–protein interactions. It is, thus, hypothesized to function as a receptor for its IRV companion partners, GLUT4 and IRAP [39].

IRVs contain minor, but functionally relevant, components, including transferrin receptors (TfR) and mannose-6-phosphate receptors (M6PR), along with various other known membrane recycling proteins [34,35]. Although the precise behavior of these minor proteins remains unclear, their presence may explain the known insulin-responsive translocation of the TfR and M6PR to the plasma membrane in adipocytes [41].

Finally, IRVs also harbor a variety of proteins that function as vesicle-soluble NSF attachment protein receptors (v-SNAREs) during membrane fusion, including vesicle-associated membrane protein 2 (VAMP2) and syntaxin-6 (Stx6) [40,42]. Other proteins may also be present as minor components, although their roles are yet to be elucidated [37], including tumor suppressor candidate 5 (TUSC5; also known as TRARG1), which has been identified as a novel GSV-associated protein, although its precise role in vesicle trafficking has not been clarified yet [48,49]. Many other proteins are also enriched in GSVs [50] and, upon insulin stimulation, translocate to the plasma membrane, while being depleted from the intracellular GSV pool. This regulated vesicle trafficking system provides a unique mechanism for keeping specific proteins functionally inactive within a sequestered compartment, enabling their mobilization to the cell surface in response to a specific type of extracellular stimulation such as insulin [38].

The formation of IRVs follows the general principles of membrane vesicle biogenesis, involving the recruitment of protein coats through adaptor proteins to a specific site on donor membranes. These adaptor proteins establish multiple interactions with the cytoplasmic domains of cargo proteins, the small GTPase Arf-GTP, and phosphatidylinositol phosphates [51]. Specifically, IRV formation on intracellular donor membranes requires clathrin-coated vesicles and Golgi-localized, Arf-binding (GGA) adaptor proteins [52,53]. Among the IRV components, sortilin is the only one currently known to directly interact with GGA adaptors via a conserved DXXLL GGA-binding motif [54]. This unique feature enables sortilin to act as a transmembrane scaffold, bridging key luminal cargo proteins, including GLUT4, IRAP, and LRP1, to a cohesive large oligomeric complex. Through this mechanism, these proteins may be co-packaged into budding IRVs via the GGA-dependent vesicle formation pathway [55]. However, GGA adaptors alone may not be sufficient to produce IRVs. Arf GAP with coiled coil, ankyrin repeat, and PH domain 1 (ACAP1) is another adaptor protein critical for IRV biogenesis. It interacts directly with the central cytoplasmic loop of GLUT4 and recruits clathrin to IRV budding sites [56]. This suggests that IRV formation may require the coordinated action of both GGA and ACAP1. Moreover, adaptor proteins (APs), such as AP1 and AP3, have also been implicated in this process, although their precise roles remain less defined [32,33]. The interplay between these different adaptor proteins is an area that warrants further investigations. Notably, both GGA and ACAP1 are known to interact with Arf6, a small GTPase involved in membrane trafficking, recycling, and secretion across various cell types. In 3T3-L1 adipocytes, Arf6 significantly colocalizes with GLUT4 at perinuclear donor membranes. Importantly, Arf6 knockdown in these cells completely obstructs insulin-stimulated GLUT4 translocation, emphasizing its essential role in IRV trafficking [56]. Although the exact phosphatidylinositol phosphate species involved in IRV formation has not yet been fully identified, recent evidence suggests that phosphatidylcholine inositol 4-phosphate (PI4P) may play a key role. PI4P has been shown to interact with GGA adaptors and facilitate their recruitment to the TGN [57,58]. Given that the TGN is considered a major donor site for IRV formation, PI4P emerges as a strong candidate in regard to this mechanism [37]. This raises the next critical question: How are the newly synthesized IRV proteins selectively targeted to the donor membranes for incorporation into IRVs?

Unlike most membrane proteins, newly synthesized GLUT4 and IRAP are not targeted directly to the plasma membrane; instead, they reach the insulin-responsive compartment within 6–9 h after synthesis [53,59,60]. Evidence suggests that these proteins bypass the plasma membrane and traffic directly from the secretory pathway to IRVs [37]. This precise targeting relies on specific cytoplasmic motifs/signals within cargo proteins. For GLUT4, these signals are located in the N-terminus and the large central loop [61], while, for IRAP, a dileucine motif at amino acids 76–77 is critical [59]. Notably, these targeting signals are distinct from those used by recycling proteins [62]. The targeting of newly synthesized GLUT4 and IRAP to the IRV compartment also requires GGA adaptor proteins [53,59]. However, it remains unclear whether these newly synthesized proteins integrate into pre-existing donor membranes along with recycling cargo or whether they are directed to IRVs via an alternative route [37]. This remains an open question in regard to understanding IRV biogenesis and trafficking.

## 3. Regulation of GSV Translocation by Insulin

Even though the translocation of GLUT4 to the plasma membrane depends on the interplay between insulin signalling and vesicle trafficking pathways, many steps in this process are not directly regulated by insulin. A key challenge is determining which steps are insulin sensitive and which have the most impact on glucose uptake. The mobilization of GSVs is tightly regulated and does not involve membrane budding or sorting. Instead, evidence suggests that TUG cleavage plays a regulatory role (discussed below). Insulin also promotes vesicle fusion at the plasma membrane [63], but the initial release of GSVs from their intracellular compartment must occur first. Notably, the primary insulin-regulated step may not be the one most impacted by insulin resistance. Therefore, mapping the intracellular pathway of GLUT4 and identifying the key insulin-regulated sites disrupted by insulin resistance is critical for understanding both normal physiology and disease [38]. Below, we explore how insulin signalling influences GLUT4 trafficking [64].

In response to postprandial hyperglycemia, pancreatic β-cells release insulin, which binds to its receptor on the cell surface of target cells. This binding activates multiple intracellular signalling pathways that drive the translocation of GSVs to the plasma membrane, enabling glucose uptake. GSV movement is orchestrated through several steps: trafficking on microtubules and actin filaments, tethering to the plasma membrane via the exocyst complex, and fusion with the plasma membrane, mediated by SNARE proteins and their associated regulators [64,65].

Upon insulin stimulation, two major signalling pathways are activated. The canonical pathway involves phosphatidylinositol-3-kinases (PI3K) and protein kinase B/AKT, and an additional wortmannin-insensitive pathway, mediated by TC10α, a PDZ protein interacting specifically with TC10 (PIST), and TUG. Both pathways contribute to GLUT4 trafficking, but they act through distinct molecular processes. The AKT pathway facilitates the recycling of GLUT4 from endosomes to the plasma membrane, especially during prolonged insulin stimulation [38]. At the membrane, the VAMP2–Synaptosome Associated Protein 23 (SNAP23)–Syntaxin-3/4 SNARE complex facilitates vesicle fusion. This process is regulated by insulin-stimulated phosphorylation of the SNARE regulator, Mammalian Homologue of UNC-18 (Munc18c), along with the involvement of Munc13 and double C2-like domain (DOC2C) proteins [66].

Mechanistically, the AKT pathway begins with tyrosine phosphorylation of insulin receptor substrate proteins (IRS1), which recruits PI3K to produce the phosphatidylinositol-3,4,5-triphosphate (PI3P) at the inner leaflet of the plasma membrane. PI3P then recruits and activates AKT through dual phosphorylation at Ser473 (or Ser474 in AKT2) via mTORC2 and, at Thr308 (or Thr309 in AKT2), via phosphoinositide-dependent protein kinase 1 (PDK1) [67,68]. AKT subsequently phosphorylates AS160/TBC1D4 that regulates GLUT4 vesicle dynamics [69]. The phosphorylation of AS160 suppresses its GAP activity toward Rab proteins, primarily Rab10 in adipocytes and Rab8A in muscle cells. This process, activates Rab proteins, promotes GSV mobilization and their subsequent fusion with the plasma membrane [25,26,50,70]. As mentioned previously, AS160 colocalizes with GSVs via its interaction with IRAP, emphasizing the specificity of this pathway regarding the insulin-responsive GLUT4 vesicle pool [45,46]. However, the precise role of AS160 in GLUT4 trafficking remains partially unresolved, as its knockdown does not fully block insulin-stimulated translocation, suggesting it may also contribute to GSV biogenesis [39].

Accumulating evidence suggests that vesicles bearing Rab10 fuse directly with the plasma membrane, while the internalization of GLUT4 post-release into endosomes likely occurs through general endocytosis rather than through a specialized recycling process. [25,26]. Identifying the precise effectors of activated Rab10 or Rab8A may clarify the identity of the target membranes involved in this fusion process. Although several downstream effectors, such as Sec16A [71], and the myristoylated alanine-rich C-kinase substrate (MARCKS) [72], have been implicated in this process, their role in GSV trafficking remains unclear. Rab10 has also been linked to lysosome–plasma membrane fusion [73], while Rab8A may promote GSV exocytosis in muscle cells via its interaction with myosin-Va [74].

Recent studies have identified a small GTPase as a key downstream effector of AKT, Rac1, that was found to be required for insulin-stimulated glucose uptake in both skeletal muscle and adipose tissue cells. Rac1 acts downstream of AKT to promote GLUT4 translocation to the plasma membrane in a RalA-dependent manner [75,76].

In parallel, insulin stimulates the TC10α pathway, a member of the Rho family GTPase, which also plays a role in GSV trafficking, particularly in adipocytes [77,78]. Upstream mediators, such as CAP and Cbl, target the transient activation of TC10α, contributing to actin remodeling in perinuclear and cortical regions. TC10α also interacts with COPI coat proteins to regulate actin polymerization on transport vesicles [79,80]. In addition, TC10α signalling is associated with the TUG pathway via PIST, a protein that directly binds and modulates TUG cleavage, further linking cytoskeletal dynamics to GSV mobilization [81,82].

Notably, AKT activation alone is sufficient to promote GLUT4 translocation in a manner similar to insulin stimulation [83,84]. However, this does not negate the contribution of alternative regulatory mechanisms, including those involving protein kinase C (PKC), TC10α, and c-Cbl, although their exact significance in the broader context of GLUT4 trafficking remains a topic of ongoing research and debate [64]. Understanding how insulin signalling integrates the molecular mechanisms of GSV translocation is essential for determining the steps affected in insulin resistance, which may facilitate the development of novel targeted therapies.

## 4. Functional Role of TUG in GSV Retention and Insulin Response

Once the IRVs are formed, they are efficiently sequestered within the cell under basal (insulin-free) conditions. While the exact mechanism of this intracellular sequestration is still not fully understood, a leading hypothesis postulates that IRVs are recycled through the TGN, possibly bypassing endosomal fusion [20,23,33], facilitating their sequestration from both endosomes and the plasma membrane [85,86,87].

A crucial element in regard to this retention mechanism is the cytosolic tethering protein, TUG, which has been identified in a functional screening as a selective regulator of GLUT4 trafficking [85,86,87]. Encoded by the *ASPSCR1* gene, TUG specifically binds GLUT4, anchoring it within the IRV pool, without affecting conventional endosomal pathways. Early studies suggested that TUG tethers GLUT4-containing vesicles to a perinuclear anchoring site, likely near the endoplasmic reticulum (ER)–Golgi intermediate compartment (ERGIC), and insulin stimulation triggers the release of this tether, allowing GSVs to mobilize to the plasma membrane [38,85,87].

TUG exclusively binds to GLUT4, but not to GLUT1, and colocalizes with non-endosomal GLUT4 in 3T3-L1 adipocytes. The interaction is thought to occur via the extensive intracellular loop of GLUT4 [87], and possibly through the N-terminus region as well, both of which are essential for insulin-responsive trafficking [61].

The specificity of the TUG–GLUT4 interaction supports the view that TUG regulates GLUT4 in IRVs rather than in other compartments. TUG’s N-terminal and central domain bind GLUT4, while the C-terminal domain, although not required for this binding, is essential for GLUT4 retention. It is proposed that the C-terminal domain of TUG interacts with an unidentified anchoring protein(s), possibly Golgin-160, that may maintain GLUT4 within the intracellular cycle that defines IRV sequestration [86], especially as the overexpression of TUG has been shown to enlarge the IRV pool [37,86,87]. Additional support for this model comes from the experiments using a dominant negative C-terminal TUG fragment (UBX-Cter), which interferes with GLUT4 retention [87]. The expression of UBX-Cter disrupts endogenous TUG anchoring by competitively binding the retention site, causing GLUT4 to redistribute to the plasma membrane, similar in magnitude to insulin stimulation. This effect occurs without altering the transferrin receptor distribution, indicating specificity for GLUT4 [86,87]. Moreover, UBX-Cter expression redirects non-plasma membrane GLUT4 to endosomes, while TUG overexpression causes GLUT4 to accumulate in non-endosomal vesicles, possibly representing IRVs [37,86,87]. In contrast, total internal reflection fluorescence (TIRF) microscopy has shown that TUG depletion in 3T3-L1 adipocytes enhances GSV exocytosis to levels comparable with insulin-stimulated control cells [25]. Notably, this approach differentiates GSVs from endosomes based on the vesicle size and kinetics. This study also revealed that insulin not only stimulates GSV release from storage, but also alters the recycling route, so that after prolonged stimulation, GSV cargoes are redirected through endosomes, before returning to the plasma membrane. This regulation appears to involve AS160 acting on Rab14 [26].

These findings underscore TUG’s pivotal role in IRV retention and support the model that insulin action targets this step as a major regulatory step [37]. If IRVs are retained via TGN-associated cycling, their components may enter this loop either directly from the secretory pathway or through retrograde transport from endosomes [37]. TUG recruitment to GLUT4-containing membranes is likely critical for their sequestration, possibly by facilitating fusion events at the TGN. Thus, the sequential actions of sortilin, responsible for cargo selection, and TUG, responsible for vesicle retention, may together confer the specificity and stability of the IRV compartment. Given that sortilin expression increases during adipocyte differentiation, this model may also explain the tissue specificity of IRV formation and regulation [55].

A critical insight into the regulation of GLUT4 trafficking by TUG was the discovery that TUG undergoes a site-specific proteolytic cleavage at the peptide bond linking residues 164–165, within the full 550-residue protein (Figure 2) [82]. This cleavage results in two distinct fragments: an 18 kDa N-terminal product, which remains associated with GLUT4-containing GSVs, and a 42 kDa C-terminal product, which interacts with Golgi/ERGIC-associated proteins. This specific cleavage event physically separates the domain that tethers GSVs from the domain that anchors them near the Golgi/ERGIC, thereby releasing the vesicles for insulin-stimulated translocation.

Importantly, insulin stimulation significantly increases the generation of these cleavage products, supporting a model in which the cleavage of TUG is a regulated and a rate-limiting step occurs in regard to GSV mobilization [37,81]. Functional relevance is underscored by the observation that a cleavage-resistant TUG mutant failed to restore GLUT4 translocation and insulin-responsive glucose uptake in TUG-deficient 3T3-L1 adipocytes [67]. These findings clearly demonstrate that TUG cleavage is not merely a by-product, but a critical regulatory mechanism through which insulin initiates GSV trafficking toward the plasma membrane. Understanding this process is, thus, essential for elucidating the physiological role of GSVs in glucose homeostasis.

The regulation of TUG cleavage is mediated through insulin signalling via the small GTPase, TC10α, and its effector protein, PIST, which directly binds to TUG [77,82,88,89]. Under basal conditions, PIST inhibits TUG cleavage, maintaining GSVs in a sequestered state. However, upon insulin stimulation, GTP-bound TC10α interacts with PIST, alleviating this inhibition and, thereby, enabling the proteolytic cleavage of TUG and the mobilization of GSVs [77,81,82].

Further regulatory complexity is added through the post-translational modification of TUG by acetylation. Specifically, lysine residues near TUG’s C-terminus are acetylated, modulating its interaction with Golgin-160, a cis-Golgi matrix protein, and acyl-CoA binding domain-containing 3 (ACBD3). These interactions help stabilize TUG-bound GSVs in a primed, insulin-responsive pool near the ERGIC/Golgi region [90]. The mutation of these acetylation sites impairs GLUT4 trafficking and blocks TUG cleavage, suggesting that these residues are essential for forming a functional protein complex that permits TUG cleavage upon insulin stimulation. Moreover, NAD^+^-dependent deacetylase (Sirtuin 2, SIRT2) interacts and specifically deacetylates TUG, hence influencing insulin-stimulated TUG cleavage and glucose tolerance in vivo. In addition, the NAD^+^-dependent deacetylase, SIRT2, plays a key role by deacetylating TUG, thereby influencing its cleavage and linking redox-sensitive cellular states to insulin signaling and glucose metabolism [91]. These observations explain how changes in metabolic conditions can modulate the cellular redox environment and directly alter insulin sensitivity by affecting GLUT4 trafficking. Furthermore, TUG cleavage is a dynamic process that is tightly controlled through a multi-step regulatory process to determine the fate of GSVs. While TUG cleavage enables the physical release of vesicles, further work is still needed to fully elucidate how TUG cleavage mechanistically coordinates with motor protein recruitment and vesicle targeting of the plasma membrane.

GSVs are sequestered in the perinuclear region of adipocytes and muscle cells, spatially separated from the plasma membrane. Kinesin motor proteins mediate their translocation to the cell membrane along microtubules [92]. An attractive hypothesis postulates that insulin-stimulated TUG cleavage may serve a dual purpose: the release of GSVs from intracellular retention, and the activation of their microtubule-dependent transport toward the plasma membrane. As discussed earlier, insulin stimulation induces TUG cleavage, generating two fragments. The 18 kDa N-terminal cleavage product remains bound to the GSVs and contains tandem ubiquitin-like domains ending in a conserved diglycine motif (residues 163–164), a hallmark feature of ubiquitin-like protein modifiers [82]. This cleavage product was named accordingly TUG–Ubiquitin-Like (TUGUL), based on its ability to covalently attach to target proteins in a manner like ubiquitin. In 3T3-L1 cells, through the use of antibodies targeting the TUG N-terminus, researchers have detected full-length TUG (~60 kDa) and an additional ~130 kDa protein. Given that TUGUL is 18 kDa, this suggests that its physiological target is a protein of 110 kDa. This “tugulated” 130 kDa protein co-fractionates with GSVs and plasma membrane, suggesting its potential involvement in GLUT4 trafficking [93,94].

In 3T3-L1 adipocytes, studies have identified kinesin family member 5B (KIF5B), the 110 kDa kinesin motor, as the major TUGUL-modified protein. This finding supports earlier work demonstrating that, upon insulin stimulation, KIF5B transports GLUT4 from the perinuclear region to the plasma membrane [95]; a process shown to be wortmannin-insensitive, independent from canonical PI3K–AKT signaling [95]. The physiological significance of KIF5B was further confirmed when adipose-specific deletion of KIF5B, in mice, caused glucose intolerance and insulin resistance [96]. These results suggest that KIF5B is not only a key TUGUL-modified substrate [93], but also a central player responsible for loading GLUT4-containing GSVs onto microtubule-based transport systems in response to insulin [93].

In adipose tissue and differentiated 3T3-L1 adipocytes, mechanistic studies have identified a splice variant of the ubiquitin-specific peptidase 25 (USP25) protease, USP25m. This variant cleaves TUG and promotes GLUT4 translocation when overexpressed [93]. Notably, the interaction between USP25m and both TUG and GLUT4 is dynamic, wherein insulin stimulation dissociates USP25m from these proteins. Altogether these findings underscore a coordinated mechanism whereby insulin triggers TUG cleavage and TUGUL generation, which then modifies KIF5B to mobilize GSVs, effectively linking their release from storage to active transport toward the plasma membrane (Figure 2) [38]. However, the precise mechanism by which the insulin signal enhances USP25m activity towards TUG remains unclear; it is likely mediated through TC10α activation and the relief of PIST-dependent inhibition, but this remains to be investigated [38].

## 5. Kinetic Models of IRV Trafficking: Retention vs. Repulsion

Two main kinetic models have been proposed to describe the recycling behavior of GLUT4-containing IRVs, based on studies using antibody-tagged GLUT4 molecules, in live adipocytes, with or without insulin stimulation. In the absence of insulin, GLUT4 recycles slowly between the plasma membrane and GSVs, in contrast to the rapid and robust translocation seen upon insulin stimulation [17,20,21,23,97]. Despite inconsistent variability in the reported extent of GLUT4 plasma membrane recycling under basal conditions, possibly due to technical differences in experimental designs, a growing consensus supports the existence of two core models: the retention model and the repulsion model [64].

In regard to the retention model, GSVs are physically anchored by interacting with special tethering proteins, such as TUG and AS160/TBC1D4, which prevent their mobilization to the plasma membrane [34,45,86,87]. These interactions likely keep GLUT4 sequestered in the perinuclear region or associated compartments, limiting its basal surface exposure.

In contrast, the repulsion model proposes that GSVs continuously approach and interact with the plasma membrane, regardless of insulin availability. However, in the absence of insulin, they fail to fuse with the plasma membrane. This model is supported by TIRF microscopy, which has revealed GLUT4-containing vesicles localized within 150 nm of the plasma membrane, a region termed the TIRF zone. In the absence of insulin, GLUT4-positive compartments accumulate beneath the membrane and may become stationary but rarely fuse with it [98,99,100]. This behavior resembles docking without fusion, aligning with the concept of membrane “repulsion” under basal conditions [64].

The repulsion model is further supported by the observations that AS160/TBC1D4 is highly phosphorylated and present at the plasma membrane following insulin stimulation [101]. This could indicate a functional role for AS160/TBC1D4 in modulating Rab activity, necessary for the GSVs’ final docking and fusion events. Moreover, insulin signalling appears to regulate both the mobilization of IRVs and their fusion at the plasma membrane, although not all these steps are AKT–PI3K dependent. In fact, several studies have shown that IRV mobilization may occur independently of PI3K/AKT signaling [95,102,103]. This principle is consistent with findings in insulin-resistant states, where GLUT4 translocation is impaired even when AKT signalling to AS160/TBC1D4 remains intact [104,105]. In this context, TC10α signalling may play an early role in IRV activation [77], whereas AKT may be more critical for later stages of GSV docking and fusion [106].

One important unresolved question is whether endocytosed GLUT4 re-enters the IRV pool during basal conditions or is redirected to alternative recycling pathways. It is generally believed that in unstimulated cells endocytosed GLUT4 must be sorted into IRVs to ensure proper sequestration. Upon insulin stimulation, these IRVs are mobilized to deliver GLUT4 to the plasma membrane. Insulin is also known to modulate the endocytic pathway [18], dephosphorylates syntaxin-16 [107], and regulates Rab proteins via AS160/TBC1D4 phosphorylation [69], suggesting multi-level control over both the trafficking and re-sequestration of GLUT4. Notably, prolonged insulin stimulation may reroute internalized GLUT4 directly from endosomes back to the plasma membrane, bypassing the IRV compartment entirely [37]. This adaptive response could help maintain surface GLUT4 levels during sustained insulin signalling, although more evidence is needed to confirm the involvement of this pathway. In conclusion, both the retention and repulsion models contribute valuable insights into the dynamics of GSV trafficking. However, the relative contribution of each mechanism remains ambiguous, primarily due to limitations in the current imaging resolution that make it difficult to distinguish GSVs and endosomes [64]. Future advancements in live-cell super-resolution microscopy will be critical to further explore the complex spatiotemporal regulation of GLUT4 trafficking.

## 6. The Lifecycle of IRVs After Insulin Stimulation

In response to insulin stimulation, GSVs are released from their perinuclear sequestration, post-TUG cleavage, which initiates their translocation to the plasma membrane [81,108]. Before fusion, GSVs engage in three sequential interactions with the plasma membrane, including approach, tethering, and docking [64]. Some GSVs may also be recycled through intracellular compartments, such as the TGN, in a Rab31-dependent manner [109].

### 6.1. Approach

Analogous to other secretory vesicles, GSVs utilize cytoskeletal motors to reach the plasma membrane. Short-range transport near the membrane is driven by actin-based myosin, notably MYO1C and MYO5A [74,110], while long-range transport from the perinuclear region is driven by microtubule-dependent kinesins (e.g., KIF5B) [95,111].

In adipocytes, MYO1C interacts with the small GTPase, RalA, which is localized within GSV membranes [78]. Activated GTP-bound RalA promotes GSV targeting through binding to exocyst subunits, SEC5 and EXO84. They are part of the eight-subunit exocyst complex (Sec3, Sec5, Sec6, Sec8, Sec10, Sec15, Sec4, and Exo70), which stabilize GSVs at the plasma membrane. Insulin-induced activation of AKT and the small GTPase, TC10α, as well as the lipid raft-localized synapse-associated protein 97 (SAP97), contribute to the docking and stabilization in regard to the plasma membrane.

Following this docking phase, PKC-mediated phosphorylation of SEC5 disturbs the interaction between RalA and the exocyst complex, effectively disengaging GSVs from the docking apparatus and priming them for fusion with the plasma membrane. While MYO1C’s role is well-established in muscle GSV trafficking [110,112], the conservation of the role of RalA and the exocyst complex, across tissues, remains under investigation. Usually, the cytoskeleton provides a pathway used by vesicles to reach the plasma membrane. However, although adipocytes lack stress fibers, they have a dense cortical actin layer underneath the plasma membrane, meaning GSVs are likely reach the cell cortex via microtubules [95].

### 6.2. Tethering: Bridging GSVs to the Plasma Membrane

Tethering bridges GSVs and plasma membranes at distances of 4–8 nm, preparing them for ternary SNARE complex-mediated fusion [113]. This process often begins at even greater distances via massive multi-subunit complexes or elongated rod-like molecules [114]. Three potential tethering systems for GSVs are the exocyst complex, cortical actin, and AS160/TBC1D4.

The exocyst complex is multi-functional, binding to phospholipids, actin, actin nucleators (e.g., Arp2/3, IQGAP1), atypical PKC, JNK, and several small GTPases (RAL, CDC42, Rho, Rab8, Rab10, Rab11, TC10α, ARF6), as well as SNAREs and their regulators, like snapin [115,116,117]. Low-resolution electron microscopy suggests that the exocyst has a ‘Y’-shaped configuration, with each tip of the ‘Y’ interacting with the plasma membrane via different small GTPases [118,119]. This exocyst complex is stabilized by insulin-induced AKT activation, the small GTPase, TC10α, signalling, in conjunction with SAP97, a protein enriched in lipid rafts. Once the GSVs are docked, PKC-mediated phosphorylation of SEC5 inhibits its interaction with RalA and the exocyst complex, thereby detaching the GSV from the docking apparatus and preparing it for fusion. The involvement of Rab10 and Rab11 on GSVs, and their interaction with the exocyst, further supports the role of this complex in anchoring vesicles to the plasma membrane [34,35].

In adipocytes, however, the actin cytoskeleton also contributes significantly to the process. Perturbation of the cortical actin, either through depolymerization or stabilization, impairs insulin-induced GLUT4 trafficking [120,121,122,123]. Although the detailed organization of cortical actin’s architecture has not been thoroughly investigated, it likely provides a more extended tethering scaffold than the exocyst, which is estimated to span distances of approximately 15 nm [118].

Myosin motors, such as MYO5 and MYO1C, have been detected on both GSVs and at the plasma membrane. These motors were believed to facilitate vesicle movement and positioning by interacting with actin filaments as mechanical tethers [124,125,126]. In addition, AS160/TBC1D4 (AS160) functions with tether-like properties. It forms homo-oligomers [127], binds GLUT4-conaining vesicles directly [34], and is associated with the plasma membrane [101]. Although, the molecular contents of these interactions remain unidentified, AS160/TBC1D4 is known to interact with various Rab GTPases [128], at least three of which also engage with the exocyst complex and MYO5 [116,129,130]. Thus, TBC1D4 plays a pivotal role as a coordinator, linking the Rab proteins, the actin cytoskeleton, and the exocyst machinery [64].

### 6.3. Docking and Fusion

As insulin initiates the final stage of GSV exocytosis, vesicles approach the plasma membrane for fusion, a tightly orchestrated molecular event, involving multiple protein partners. Central to this process is the formation of a SNARE ternary complex and the participation of Sec1/Munc18-like (SM) proteins. These proteins are believed to be constitutively active in vitro and require tight regulation in vivo to ensure insulin-dependent specificity and temporal precision [131]. In this context, the primary SNARE players are the t-SNAREs (syntaxin-4 and SNAP23) and the v-SNARE (VAMP2), while Munc18c serves as the SM regulatory protein [33,132]. Together, they mediate the membrane fusion of GSVs with the plasma membrane through a zipper-like mechanism, where the SNARE motifs coil from the N- to C-terminal ends, bringing the two membranes into close proximity and overcoming the energy barrier for fusion [133]. The recruitment of SNARE accessory proteins modulates this process. Doc2b, Synip, and Munc18c are recruited to the plasma membrane and regulate the assembly and function of the SNARE complex [134,135,136,137,138,139,140,141,142,143,144,145,146].

Munc18c is of particular interest; it binds to the N-terminal region of syntaxin-4, possibly stabilizing it in an “open” conformation, and associates with the full SNARE complex [131,147,148]. However, the exact function of Munc18c remains unclear. While Munc18c promotes SNARE complex formation in some settings, overexpression studies of adipocytes suggest an inhibitory role [149]. Genetic deletion mutants of Munc18c have a limited impact on GLUT4 trafficking [150], yet its phosphorylation at Y521 in response to insulin appears crucial for GSV exocytosis [143,151]. This phosphorylation occurs within a disordered, surface-exposed domain, potentially modulating its interaction with other fusion regulators [64].

Calcium signaling is also implicated in GLUT4 trafficking. C2 domain-containing proteins like extended synaptotagmin-1 (ESYT1) and double C2-like domain beta (DOC2b) have potential implications in regard to the GLUT4 trafficking fusion site. In muscle cells, insulin elevates Ca^2+^ concentrations beneath the membrane [152], and the chelation of Ca^2+^ with 1,2-bis(2-aminophenoxy)ethane-N,N,N′,N′-tetraacetic acid (BAPTA) inhibits translocation in adipocytes [153]. ESYT1 is phosphorylated by cyclin dependent kinase 5 (CDK5), localized to the plasma membrane [154], while DOC2b translocates to the membrane and binds syntaxin-4 in a Ca^2+^-dependent manner [155]. DOC2b acts as a positive modulator of SNARE assembly and enhances fusion by inducing membrane curvature upon insertion [154,155]. These proteins are concentrated at fusion sites, where they interact with both GSV and plasma membrane SNAREs, thereby physically linking the bilayers to facilitate fusion [64,156]. Synip and Munc18c, which both bind syntaxin-4, show negative regulation when overexpressed, but under physiological conditions, Munc18c seems to be a limiting component for GLUT4 fusion [139,140,141,145,149], because knock-out (KO) mice exhibit impaired GLUT4 accumulation in sarcolemmal and t-tubule membranes of skeletal muscle [144]. Upon syntaxin-4 activation, it contributes to the development of the SNARE complex and GSV fusion. Notably, both Doc2b [157] and syntaxin-4 [158,159,160] engage with cytoskeletal elements, which may be essential for the mechanism facilitating GSV fusion [111].

In addition to the importance of translocation and fusion of GSVs at the plasma membrane for increasing GLUT4 availability in the plasma membrane, recent super-resolution studies have identified an additional insulin-regulated step involved in this process. Advanced imaging has allowed researchers to demonstrate that GLUT4 does not remain clustered at fusion sites but rather diffuses across the membrane in an insulin-regulated manner. This change in the spatial distribution of the transporter provides an additional regulation point that may enhance glucose uptake. This recent finding increases our understanding of GLUT4 regulation beyond vesicle fusion and may provide new targets for the treatment of insulin resistance [161].

### 6.4. Reconstitution of IRVs After Insulin Stimulation

Every cycle of insulin stimulation results in the translocation of the entire complement of IRV proteins, including GLUT4, IRAP, sortilin, and LRP1, to the plasma membrane. Because kiss-and-run fusion events are relatively rare, these proteins are likely internalized at varying rates post-insertion into the plasma membrane. This situation requires a robust and high-fidelity mechanism to reassemble functional IRVs from their constituent components after each cycle of exocytosis, namely GLUT4, IRAP, sortilin, and LRP1, to the plasma membrane. Over the lifespan of a skeletal muscle fiber or adipocyte, this cycle is repeated hundreds to thousands of times. Any defect in this reassembly process can lead to reductions in IRV abundance or insulin sensitivity, thereby contributing to insulin resistance [162].

This raises an important question: How are IRVs regenerated after insulin-stimulated depletion? During intracellular trafficking (Figure 3), GLUT4 moves through multiple compartments, including early and sorting endosomes, recycling endosomes, and/or TGNs [39].

Once insulin signalling is eliminated, the cell initiates a retrieval process to internalize GLUT4 from the plasma membrane. To maintain glucose homeostasis, endocytosis occurs through two principal pathways: clathrin-mediated endocytosis and cholesterol-dependent endocytosis [163]. Historically, clathrin-mediated endocytosis is considered the primary endocytic pathway in skeletal muscle cells. However, evidence from the studies of rat L6 myoblasts has revealed that depleting cholesterol significantly inhibits GLUT4 internalization and endocytosis, highlighting an essential role of cholesterol-rich microdomains in this process [163]. In classical clathrin-mediated endocytosis, adaptor protein-2 (AP2) binds directly to GLUT4, recruiting clathrin to the inner leaflet of the plasma membrane. This interaction mediates the formation of clathrin-coated vesicles, which bud out from the plasma membrane with the help of the GTPase dynamin [164]. Newly formed vesicles/endosomes, now containing GLUT4, are handed off to the microtubule-based motor protein dynein, which, through the interaction with Rab5 on the vesicle surface, directs their retrograde movement toward the cell interior [165].

Following endocytosis, IRV proteins are internalized from the plasma membrane into sorting endosomes, where the fate of GLUT4 is decided. Some are either directed towards breakdown due to lysosomal degradation, while a significant portion is reserved for recycling and eventually reach a syntaxin-6/16-positive perinuclear compartment, which may represent a sub-domain of TGNs and recycling endosomes [32,33]. In human skeletal muscle, the retrograde transport of GLUT4 to TGNs is essential and appears to involve syntaxin-10 [166] and is facilitated by the clathrin heavy chain, CHC22 isoform [167]. Biochemically, the perinuclear compartment consists of a mixture of small IRVs and large donor membranes, maintained in a dynamic equilibrium [23,24,55,168,169]. While conventional fluorescence microscopy cannot resolve these structures, biochemical fractionation techniques enable their separation and study [168]. The reconstitution of GSVs is thought to involve budding and fusion cycles between IRVs and their donor membranes [23,55]. Both biosynthetic and recycling pathways must converge here to deliver cargo for the formation of new IRVs. Notably, newly synthesized GLUT4 and IRAP do not traverse to the plasma membrane directly. Instead, they are incorporated into GSVs within 6–9 h post-translation on the endoplasmic reticulum (ER) [53,59,60]. Moreover, IRV proteins recycled from the plasma membrane pass through sorting and recycling endosomes and, particularly, the TGN, before being reloaded into nascent IRVs [15,31,35]. Thus, the formation of functional GSVs depends on the convergence of biosynthetic and endocytic pathways.

Sorting of GLUT4 into GSVs is mediated by unique cytosolic signals on cargo proteins [55,168,170,171]. In GLUT4, such motifs are found in the N-terminal and the central cytoplasmic loop [61]. The N-terminal phenylalanine-based motif is required for GSV targeting and internalization, while the C-terminus guides GLUT4 to the perinuclear donor compartment without ensuring IRV entry [18,61,170,172]. GLUT4 ubiquitination also facilitates its packaging into GSVs via interactions with GGA proteins [173]. In human muscle and adipose tissues, but not in mice, CHC22 mediates the regulated trafficking of GLUT4, implicated in GSV formation at an as-yet unidentified membrane site [38,166,167,174]. For IRAP, the cytosolic N-terminus directs the protein to the donor membranes, but not into IRVs themselves [168]. A critical dileucine motif (positions 76 and 77) is essential for insulin responsiveness [59]. These motifs are distinct from motifs mediating generic recycling [62]. These motifs likely interact with proteins that are peripherally associated with GSVs, such as TUG and AS160/TBC1D4, both of which bind IRAP and participate in GSV trafficking [39,47,86,87,175].

The signals within the cytoplasmic tails of the principal IRV components, GLUT4, IRAP, and sortilin, as well as LRP1, are crucial for their localization to the IRV donor compartment yet are not sufficient for their incorporation into IRVs. This suggests that luminal domains are vital for protein localization in the subsequent compartment. Notably, GLUT4, IRAP, sortilin, and LRP1, can engage with one another through their luminal domains [35,55,168,170,171,176,177] to form an oligomeric complex that can be inserted into the IRVs as a single entity with the help of GGA and ACAP1 [37,56]. Notably, sortilin, a Vps10p homolog, expression is essential and sufficient for GSV formation in adipocytes [55,171,178]. It serves as a major scaffold linking luminal cargo complexes to cytosolic adaptors, such as GGA proteins, which recognize the sortilin’s DXXLL motif [54], while ACAP1 binds to the central loop of GLUT4 [37,56]. These adapters are recruited to the GSVs by Arf6 and PI4P, ensuring the precise assembly of the budding machinery at donor membranes [55,179,180,181,182,183].

Thus, the combination of luminal and cytosolic signals guides IRV reconstitution. These interactions are also functionally interdependent: knockout or knockdown studies of one component (e.g., GLUT4, IRAP, LRP1, or sortilin) reduce the abundance of others post-transcriptionally, and disrupt IRV formation [35,55]. Nevertheless, IRVs may likely still be generated, albeit with reduced efficiency, indicating redundancy among the core components [184]. Notably, proteins that are not properly sorted into IRVs are degraded [47,55,180,181,182].

Taken together, these observations support a model wherein the interaction within the luminal domains of the core cargo proteins mediates self-assembly and the reconstitution of IRVs, which are further stabilized by cytosolic sorting signals and budded from donor membranes via adaptor-mediated mechanisms. These cycles of sorting, assembly, and budding ensure the efficient and accurate restoration of the insulin-responsive GLUT4 trafficking pathway after each round of stimulation.

## 7. Involvement of Actin Cytoskeleton Remodeling in GSV Trafficking

The mobilization of GSVs from intracellular compartments toward the plasma membrane is not a passive process; it requires a precisely choreographed interplay between signaling pathways and dynamic cytoskeletal elements [95]. In skeletal muscle cells and adipocytes, both actin- and microtubule-based cytoskeletal networks contribute to this regulation [95,185,186].

Actin cytoskeleton, particularly filamentous actin (F-actin), plays a vital role in the final stages of GLUT4 vesicle mobilization. Experimental depolymerization of F-actin using Latrunculin B shows a dose- and time-dependent reduction in insulin-stimulated glucose uptake in skeletal muscle cells, suggesting that an intact F-actin network is essential for proper GLUT4 translocation [187]. In L6 muscle cells, Myo1c knockdown disrupts actin filaments, while its overexpression results in the immobilization of GSVs near the plasma membrane, as visualized by TIRF microscopy [110]. These findings support the model in which vesicular Myo1c anchors GSVs to F-actin, facilitating their final approach and fusion with the plasma membrane [110]. Notably, insulin stimulation promotes F-actin remodeling in skeletal muscles, facilitating GSV translocation to the plasma membrane [122,188]. The p21-activated kinase, PAK1, regulates both F-actin polymerization and depolymerization during insulin-stimulated GLUT4 translocation and glucose uptake in L6-myoblasts and myotubes [189,190]. The key player in this process is the Arp2/3 complex, which consists of seven subunits (ARP2, ARP3, and ARPC1-5), which promotes the nucleation of a branched actin filament network [191]. The knockdown of ARP3 or ARPC2 subunits in skeletal muscle cells inhibited insulin-induced F-actin remodeling and GLUT4 trafficking [191].

PAK1 directly phosphorylates ARPC1 (P41-ARC), thereby activating the ARP2/3 complex [190,192,193]. Although the ARP2/3 complex can initiate branched actin filament nucleation, it requires the involvement of nucleation-promoting factors (NPFs), such as the neural Wiskott–Aldrich syndrome protein (N-WASP), WAVE, and cortactin, to function efficiently. Among these, N-WASP has been directly associated with insulin-stimulated GLUT4 translocation [190,194]. While cortactin plays a supportive role, it is considered a relatively weak Arp2/3 activator and requires a strong NPF, such as N-WASP, to initiate actin polymerization.

Interestingly, insulin signaling connects the actin cytoskeleton with the SNARE fusion machinery. The inhibition of N-WASP or PAK1 activity suppresses syntaxin-4 activation, suggesting that cytoskeletal dynamics directly influence GSV docking and fusion [190]. In support of this process, foldrin, a non-erythroid spectrin, that stabilizes actin filaments, interacts with the t-SNARE protein syntaxin-4 in response to insulin and is implicated in GSV docking and fusion in adipocytes [159]. Moreover, the calpain-mediated cleavage of foldrin has been observed in Duchenne muscular dystrophy and T2D, linking actin remodeling defects to metabolic diseases [111,195,196].

Actin-severing proteins, like cofilin and gelsolin, are also important regulators [191]. PAK1 knockout mice failed to dephosphorylate phospho-cofilin in skeletal muscle tissue following insulin stimulation, suggesting a role for PAK1 upstream of cortactin in actin disassembly [189]. The two actin depolymerizing factors (ADFs), gelsolin and tropomodulin3 (Tmod3), have a potential role in PAK1-mediated actin depolymerization. Gelsolin is ubiquitously expressed [197,198], and enhances syntaxin-4-mediated fusion [199]. Tmod3, on the other hand, is phosphorylated by AKT2 at Ser71 and is essential for insulin-induced actin remodeling in 3T3-L1 adipocytes. Together, these findings reveal a tightly regulated network of actin-modifying factors that coordinate insulin-stimulated F-actin remodeling, enabling effective GSV exocytosis in skeletal muscle and adipose tissues [111].

## 8. Mechanistic Links Between GSV Dysfunction and the Development of Insulin Resistance

Maintaining systemic blood glucose homeostasis after a meal requires rapid insulin secretion from pancreatic β-cells and a coordinated response in insulin-sensitive organs. Insulin sensitivity, the capacity to normalize blood glucose at a given insulin concentration, is a complex, system-wide property, involving reduced hepatic glucose production, enhanced glucose uptake by muscle and adipose tissues, the inhibition of adipose lipolysis, and the promotion of lipid storage in the liver and fat deposits [200]. In obesity and T2D, insulin resistance develops, marked by impaired insulin-stimulated glucose uptake and persistent hepatic gluconeogenesis [201,202]. It is noteworthy to mention that overt T2D develops only when pancreatic β-cells fail to secrete enough insulin to compensate for this resistance [203,204].

Over the past 40 years, research has deepened our understanding of insulin resistance, yet the debate continues over its molecular origin. There are two schools of thought that attempt to mechanistically explain insulin resistance. One school argues that insulin resistance stems from defects in the proximal insulin signalling cascade. This is supported by clinical evidence from obese individuals undergoing bariatric surgery, in whom weight loss restores AKT activation, but fails to normalize glucose disposal, indicating a persistent defect downstream or parallel to AKT signalling [205]. The second school hypothesizes that insulin resistance alters the metabolic flux. However, this is challenged by findings that insulin resistance persists ex vivo, outside of systemic influences [206]. Pharmacological treatments and dietary shifts likewise fail to reverse insulin resistance, implying intrinsic cellular defects [200]. Therefore, this section will address the current evidence on molecular and cellular abnormalities that have been proposed to underlie insulin resistance.

### 8.1. Proximal Insulin Signalling: Cause or Consequence

Rare syndromes, like Donohue and Rabson–Mendenhall syndromes, caused by mutations in the insulin receptor (INSR) gene or autoantibodies against the receptor, confirm that defects in proximal signaling can induce insulin resistance [207]. Similarly, deactivating mutations of AKT and its upstream effectors induce insulin-resistant phenotypes in mice, rats, and humans [5,200,208,209,210,211]. Moreover, the association between cellular stresses and insulin resistance, due to the activation of intracellular Ser/Thr kinases, including PKCs, JNK, mTOR, and S6K, that phosphorylate and inhibit the insulin receptor and IRS proteins, forms a potential negative-feedback mechanism that impedes insulin signalling [212,213,214].

Although these molecular alterations may occur, evidence suggests that defects in proximal insulin signalling, including compromised insulin receptor function or alterations in the expression or number of receptors, are not the major factor contributing to insulin resistance in T2D, especially as insulin-binding studies have demonstrated that a small fraction (2.4%) of total insulin receptors is sufficient to elicit maximal signalling, supporting the term “spare receptor” hypothesis [215,216]. Thus, a reduction in the number of receptors or proximal signalling only causes a right shift (reduced sensitivity) in the insulin dose-response curve without affecting maximal responsiveness [216]. The combined right and downward shifts in glucose uptake seen in insulin-resistant individuals imply defects downstream of the receptor [216,217,218].

Redundancy also exists for downstream effectors: AKT2 deletion reduces AKT phosphorylation by 90% without impairing glucose uptake or protein synthesis, as residual AKT1 activity compensates for this [219,220,221,222,223]. Even when AKT phosphorylation is reduced, the downstream substrate phosphorylation remains intact, both in animal models and human T2D muscles [101,104,200,221,222,224,225,226,227,228,229,230,231].

Genetic and pharmacological models that bypass INSR/IRS signaling (e.g., platelet-derived growth factor receptor (PDGFR) overexpression) still show impaired glucose uptake in insulin-resistant muscles, refuting the role of receptor defects or IRS phosphorylation in the pathogenesis [96]. In humans, maximal insulin stimulation fails to normalize muscle glucose uptake despite normal or mildly reduced AKT activity [227,228,229,232,233,234,235,236,237,238]. Global phosphoproteomics confirms that only a minority of alterations in insulin-resistant states involve canonical signaling proteins, which mostly involve uncharacterized pathways [237,238,239,240]. Thus, while proximal insulin signaling defects may be a contributing factor, they are unlikely to be the primary drivers of insulin resistance. Importantly, unlike INSR or AKT, GLUT4 does not exhibit sparing; haploinsufficiency in mice leads to metabolic dysfunction [241]. Still, while adipose GLUT4 levels fall in T2D, skeletal muscle GLUT4 remains normal, implying that impaired trafficking, not abundance, is the critical issue [108,238].

### 8.2. Mis-Sorting of GLUT4

Prior to insulin stimulation, GLUT4 is sequestered into specialized GSVs distinct from the general secretory mechanism [49]. Mis-sorting diverts GLUT4 into non-insulin-responsive compartments, compromising translocation. Sub-fractionation of skeletal muscle or adipose tissues from insulin-resistant and T2D individuals reveals GLUT4 accumulation in denser fractions, in non-GSV compartments [242,243,244]. Mis-sorting may result from the altered expression of key trafficking proteins like sortilin and syntaxin-16 or from lipidomic changes, such as ceramide buildup. Ceramides fragment the Golgi apparatus and impair ER–Golgi trafficking, thereby inhibiting GSV formation [241,245,246,247]. In vitro, C2-ceramide disrupts GLUT4 recycling in L6 myoblasts, mimicking insulin resistance [248]

### 8.3. Impaired GSV Translocation to the Plasma Membrane

Following insulin stimulation, GSVs are mobilized to the plasma membrane, docked beneath it, and subsequently undergo SNARE-dependent fusion. TIRF microscopy showed that hyperinsulinemia-induced insulin resistance in 3T3-L1 adipocytes accumulated lower GSVs at the plasma membrane, although fusion appeared unaffected [105]. In human adipocytes, both fusion and docking are impaired [249]. Moreover, direct stochastic optical reconstruction microscopy (dSTORM) super-resolution imaging further demonstrates reduced GLUT4 dispersal across the plasma membrane in hyperinsulinemia-induced insulin resistance in 3T3-L1 adipocytes [250,251].

These defects may stem from imbalances in the SNARE complex protein or their regulators [252,253]. In diabetic rat muscle, VAMP2/3/5 and STX4 are upregulated, while VAMP5 and SNAP29 levels rise in insulin-resistant cardiac muscles. Dysregulated stoichiometry may form non-productive SNARE complexes. Munc18c, a syntaxin 4 (STX4) inhibitor, is elevated in regard to its insulin-resistant status, further altering GSV fusion [254].

### 8.4. Cytoskeletal Impairment in GLUT4 Trafficking

GSV movement is facilitated by microtubule and actin filaments. Insulin remodels both networks, enhancing microtubule polymerization and cortical actin formation [95,120,122,255,256,257,258,259,260]. In insulin-resistant models, cortical F-actin is disrupted, correlating with increased membrane cholesterol [122,248,261,262,263,264]. It is noteworthy to mention that in vivo, the role of actin in GLUT4 trafficking may be modest; the muscle-specific deletion of both b- and γ-actin isoforms in adult mice has minimal effects on insulin-stimulated glucose uptake [265,266].

Microtubule-based trafficking is more significantly affected, especially since diet-induced obesity and ceramide treatment impair microtubule polymerization and GLUT4 transport [267]. Possible mechanisms may involve altered microtubule–GSV–KIF5B interactions or disrupted microtubule dynamics, possibly through microtubule affinity-regulating kinases (MARK kinases) [239,268].

Altogether, current evidence supports the idea that the primary cause of insulin resistance is not defects in proximal insulin signaling. Instead, defective GSV sorting, trafficking, cytoskeletal dynamics, and membrane fusion are recognized as the main causes of GLUT4 dysfunction and systemic insulin resistance. Therefore, it is essential to carry out examinations beyond classical signaling pathways to fully comprehend and address the molecular origins of insulin resistance.

## 9. Outstanding Questions and Future Directions

In this review, we attempt to summarize the fundamental mechanisms of GLUT4 trafficking, highlighting its significance in maintaining glucose homeostasis in health, and its dysfunction in insulin resistance and T2D. A significant challenge that impedes our understanding of such diseases is our inability to identify the exact point(s) of dysfunction within the GLUT4 trafficking pathway in insulin resistance. Therefore, future research should be directed toward identifying these steps in both physiological and insulin-resistant contexts. This requires a thorough molecular analysis of each trafficking step, as well as its regulators. Addressing such a challenge requires advanced biochemical fractionation methodologies, high-resolution live-cell imaging, and super-resolution microscopy to resolve the complex regulation of GSV mobilization, tethering, docking, and fusion. These tools could also aid in distinguishing direct defects in GSV dynamics from compensatory responses to systemic metabolic stress.

Recently, GSVs have been considered a specialized organelle that regulates the transport and physiological functions of different proteins. This suggests that GSV trafficking defects may have other physiological and pathological consequences, in addition to insulin resistance. Therefore, elucidating the biological significance of GSV trafficking defects may point to previously unknown associations between vesicle trafficking and complex metabolic diseases.

To better understand GSV dysfunction, several questions need to be answered, including how molecular signals regulate the selective sorting of GLUT4 and other cargo proteins into IRVs versus non-insulin-responsive compartments, and which steps in the GSV trafficking process are defective in insulin resistance. Moreover, what alterations in GSV dynamics or protein composition are observed in insulin resistance? Also, identifying differences between the proteomes of healthy and insulin-resistant cells may directly identify which cellular mechanism is most affected in these cells. This may also answer the question about the extent to which GSV trafficking impacts the regulation of other proteins and may point to tissue-specific differences that may be therapeutically targeted. By clarifying the regulation and pathology of GSVs, researchers may uncover novel therapeutic targets capable of restoring insulin sensitivity and mitigating metabolic disease progression.

## Figures and Tables

**Figure 1 ijms-26-07568-f001:**
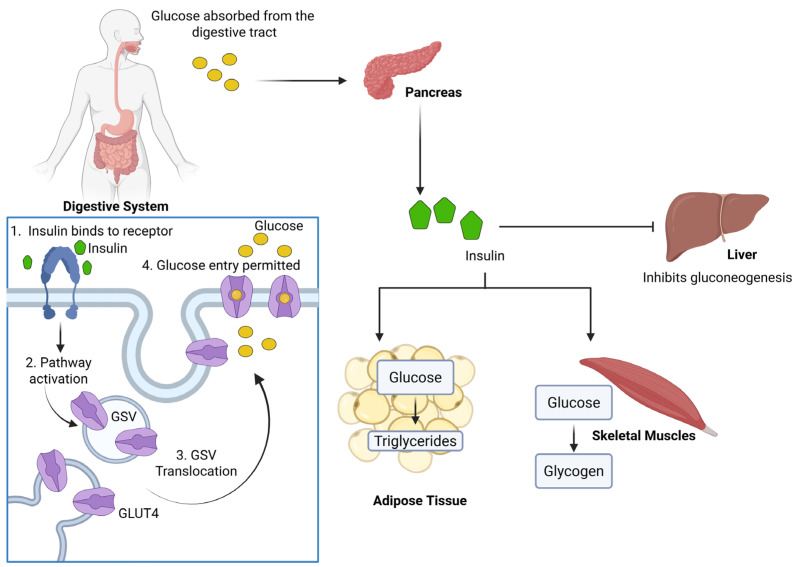
Systemic glucose homeostasis maintenance. The maintenance of systemic glucose homeostasis is a complex mechanism that relies on the regulated function of multiple organs, including the digestive system, the pancreas, liver, and muscle and adipose tissues. Following a meal, the digestive system absorbs dietary carbohydrates and amino acids into the bloodstream, which, in turn, stimulates pancreatic β-cells to release insulin, an anabolic hormone that promotes the conversion of simple nutrients into storage forms, such as glycogen, lipids, and proteins. Most glucose is conserved in the muscles in the form of glycogen. While the remaining glucose, around 10%, is taken up by adipose tissue, where it is stored in the form of triglycerides. The uptake of glucose into these cells is facilitated by the glucose transporter, GLUT4, which, in response to insulin stimulation, translocates from its intracellular sequestration to the plasma membrane, enabling glucose uptake.

**Figure 2 ijms-26-07568-f002:**
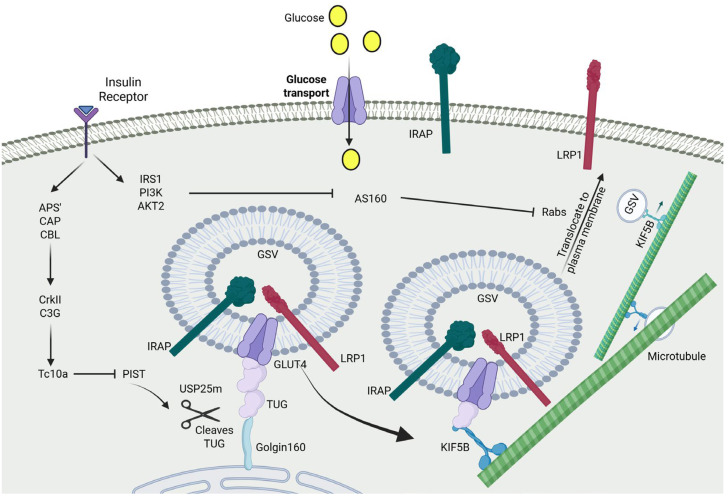
Insulin-stimulated TUG cleavage enables GSV trafficking to the plasma membrane. Insulin binding to its receptor stimulates two pathways, the PI3K–AKT and TC10α pathways. Activation of the TC10α pathway activates ubiquitin-specific peptidase 25m (USP25m)-mediated proteolysis by removing the inhibitory effect of PIST. This cleavage releases the GSVs from their intracellular sequestration through the formation of the TUG N-terminal product (TUGUL), which modifies the kinesin motor (KIF5B) and enables the translocation of the GSVs to the plasma membrane on microtubules. This allows the insertion of GLUT4, IRAP, and LRP1 into the membrane.

**Figure 3 ijms-26-07568-f003:**
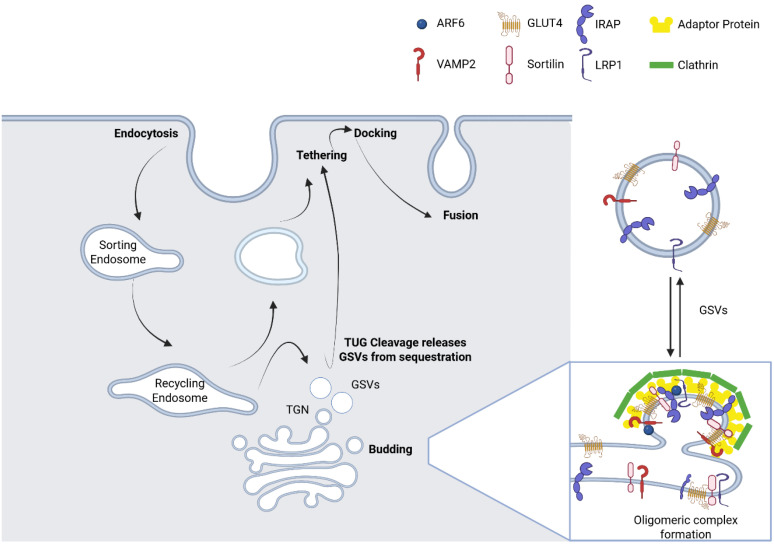
Overview of GLUT4 trafficking pathway and GSV formation. GLUT4 continuously cycles between the plasma membrane and intracellular compartments. After internalization, GLUT4 can return to the plasma membrane via two main recycling routes, namely a rapid pathway through sorting endosomes, or a slower pathway through recycling endosomes. Moreover, GLUT4 is intracellularly sequestered into GSVs that originate from recycling endosomes and/or the trans-Golgi network. In response to insulin, the rate of GSV exocytosis increases, raising plasma membrane GLUT4 levels by 10 to 30-fold. The formation of these vesicles requires the assembly of cargo proteins, including GLUT4, IRAP, LRP1, and sortilin, into oligomeric complexes within donor membranes. The budding of GSV vesicles from donor membranes is facilitated by ARF6, which recruits adaptor proteins and clathrin, enabling vesicle budding.

**Table 1 ijms-26-07568-t001:** Summary of GSV proteins and their roles in GSV dynamics processes.

Process	Proteins Involved
GSV cargo proteins	GLUT4, IRAP, Sortilin, LRP1
GSV sequestration	TUG, PIST, Golgin-160
GSV release	TC10α, USP25m
GSV translocation	AS160, RAB, TUGUL, Kinesin motor, KIF5B
GSV tethering, docking, and fusion with plasma membrane	v-SNARE VAMP2, t-SNARE syntaxin-3/4 and SNAP23, Munc18C, Mun13, DOC2, Synip
Retrograde traffic	Clathrin, Caveolin, CHC22
GSV budding	GGA, ACAP1, AP1, AP2, AP3, ARF6

## Data Availability

Not applicable.

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
