# Peer review of "GLUT4 Trafficking and Storage Vesicles: Molecular Architecture, Regulatory Networks, and Their Disruption in Insulin Resistance"

_ijms, 2025, doi:10.3390/ijms26157568_

Round 1
Reviewer 1 Report
Comments and Suggestions for Authors
- It would be appropriate to identify whether vesicles translocate to other insulin-sensitive tissues: all skeletal muscle and the liver (first paragraph on page 2). If not, describe this as an area of research.
- The term “diabetes mellitus” is now recognized solely as “diabetes” by the International Diabetes Federation (IDF).
- It would be appropriate to point out in Figure 1 that glucose is stored as glycogen in skeletal muscle to maintain the same flow of information as in other insulin-sensitive tissues.
- Place the letters "beta" and "alpha" as symbols in figure 1 and throughout the manuscript.
- Be sure to define all abbreviations the first time they appear in the document: ABX, etc.
- One fact that may support his evolutionary hypothesis (p. 5) of the reason for the "complex and heterogeneous vesicular system for regulating glucose uptake in response to insulin" is the integration of metabolism: Activation of metabolic pathways occurs from the availability of substrates, so then, anabolism of proteins, carbohydrates and lipids takes place in the cell.
- Keep case consistent when referencing Irvs, AKT, Irvs, Tug, Glut4, etc.
- It seems that the expression on page 11: “TUGUL and the Microtubule-Mediated Translocation of GSVs” is a subtitle.
- Authors should list the signaling pathways they refer to in the sentence on page 7: This binding activates multiple intracellular signalling pathways that drive the translocation of GSVs to the plasma membrane, enabling glucose uptake.
- The format of the references must be homogeneous within the references list.
Author Response
We would like to thank the reviewer for his/her kind recommendations to improve our manuscript. We would also like to notify you that several sections have been changed and modified as requested by the second reviewer.
- It would be appropriate to identify whether vesicles translocate to other insulin-sensitive tissues: all skeletal muscle and the liver (first paragraph on page 2). If not, describe this as an area of research.
We appreciate the reviewer’s thoughtful comment. However, we respectfully note that the introduction [Page 2, lines 56-59] differentiates between the roles of insulin in hepatic glucose production and glucose uptake in muscle and adipose tissues. This wording clearly differentiates between the fact that GSVs translocate in skeletal muscles and adipocytes, but not in the liver, where insulin reduces glucose output through other mechanisms.
- The term “diabetes mellitus” is now recognized solely as “diabetes” by the International Diabetes Federation (IDF).
We appreciate the reviewer’s attention to terminology. we acknowledge that the IDF recognizes the shorter term “diabetes”, therefore we deleted the term “mellitus”.
- It would be appropriate to point out in Figure 1 that glucose is stored as glycogen in skeletal muscle to maintain the same flow of information as in other insulin-sensitive tissues.
We thank the reviewer for this helpful suggestion. We modified Figure 1 to indicate that glucose is stored as glycogen in skeletal muscles. Please see Page 2.
- Place the letters "beta" and "alpha" as symbols in figure 1 and throughout the manuscript.
We thank the reviewer for the notification. “b-cells” was corrected to “β-cells” in Figure 1- legend and the other locations in the manuscript.
Be sure to define all abbreviations the first time they appear in the document: ABX, etc.
We thank the reviewer for pointing this out. All abbreviations have been defined upon first mention in the manuscript.
- One fact that may support his evolutionary hypothesis (p. 5) of the reason for the "complex and heterogeneous vesicular system for regulating glucose uptake in response to insulin" is the integration of metabolism: Activation of metabolic pathways occurs from the availability of substrates, so then, anabolism of proteins, carbohydrates and lipids takes place in the cell.
We thank the reviewer for his/her clarification. However, due to the major changes done in the manuscript as requested by the second reviewer, we deleted some perspectives regarding the heterogeneity in GSVs.
- Keep case consistent when referencing Irvs, AKT, Irvs, Tug, Glut4, etc.
We appreciate the reviewer’s attention to detail. Some formatting inconsistencies were noticed by us as well (e.g. in the capitalizations of IRVs, GSVs, etc.) as well as in the headings numbering in the version of the manuscript received after submission. We have now carefully reviewed and corrected all relevant terms to ensure consistent formatting throughout the manuscript.
- It seems that the expression on page 11: “TUGUL and the Microtubule-Mediated Translocation of GSVs” is a subtitle.
We thank the reviewer for the observation. The expression “TUGUL and the microtubule-mediated translocation of GSVs” is indeed intended as a subtitle. After the major modification, this section was masked under “Functional role of TUG in GSVs retention and insulin response”
- Authors should list the signaling pathways they refer to in the sentence on page 7: This binding activates multiple intracellular signalling pathways that drive the translocation of GSVs to the plasma membrane, enabling glucose uptake.
We thank the reviewer for this suggestion. As we describe the specific intracellular pathway in detail in the subsequent paragraphs, we have kept the sentence on page 7 general to avoid redundancy.
- The format of the references must be homogeneous within the references list.
We thank the reviewer for the notification. We corrected the references accordingly.

Reviewer 2 Report
Comments and Suggestions for Authors
The manuscript by Hana Drobiova and co-authors is a review analyzing the molecular mechanisms of insulin-dependent trafficking of glucose transporter GLUT4 in insulin-sensitive cells such as skeletal muscles and adipocytes. In these cells, the translocation of GLUT4 into the plasma membrane is carried out in a specialized type of vesicles, GLUT4 storage vesicles (GSVs). In the review, the authors consider in detail various aspects of GSV functioning, such as the biogenesis and molecular composition of GSVs, the mechanisms of their retention in the cytoplasm under basal conditions, and the signaling pathways of insulin that control GSV trafficking. The authors analyze in great detail the protein machinery mediating the sequential stages of GSVs exocytosis, including approach, tethering, docking, and fusion of GSVs with the plasma membrane. The review pays also much attention to the role of the cytoskeleton in GST translocation. Next, the review examines the various currently proposed concepts of molecular mechanisms of GSV dysfunction in insulin resistance.
Obviously, the range of problems considered in the review is highly relevant for modern molecular medicine, given the high incidence of type 2 diabetes in the human population. In the text of the manuscript, the authors demonstrated a deep knowledge of the subject under consideration and a deep fundamental analysis of the review topic.
Nevertheless, in my opinion, the manuscript is not free from a number of serious shortcomings that need to be addressed before it can be published. In fact, I think the whole paper should be re-written and re-structured. My comments for the authors are listed below.
- The quality of any review is largely determined by the choice of information selected for analysis. The translocation of GLUT4 and its disruption in insulin resistance is a hot topic on which hundreds of articles are published annually. This manuscript contains 282 references (quite a substantial number), of which 21 papers (7%) were published before 1995, 78 papers (28%) were published in the decade 1995-2004, 130 papers (46%) appeared between 2005 and 2014, and only 53 papers (19%) appeared in the last decade 2015-2025. In sum, 74% of publications analyzed in this manuscript were published 30 to 10 years ago!! In fact, this choice of information sources makes this manuscript belong more to the history of science than to current research. I understand that the basic concepts of the insulin-dependent GSV translocation were indeed mainly formulated about 20-25 years ago. But a review in a highly-rated journal should describe the current state of the art, rather than facts known for decades. In my opinion, there is no need to characterize long-known things in such detail, making hundreds of references to old and very old works. I highly recommend that the authors update the review, significantly shifting the proportion towards citation of research from the last decade.
- This manuscript is poorly structured and, as a result, very long. Many things are repeated twice in different sections. The division into chapters made in this version does not seem logical enough. I would suggest the following changes:
I think, section «Structural and Biochemical Characteristics of GSVs» can be easily merged with section “ Biogenesis of Irvs” and section “Targeting of The Newly Synthesized Irv Proteins” since in all cases authors are talking about essentially the same processes and molecules, and resident protein targeting is a part of GSV biogenesis. It would be advisable to name this section "Biogenesis and molecular composition of GSVs”. The text of this section could be significantly shorter than the three sections in the current version. Also, it would be wise to add a table with information on the composition and known functions of the proteins contained in GSVs.
Similarly, sections “Regulation By Insulin” and “Insulin Pathway Involved In Gsv Translocation” also can be merged and shortened, collectively named as “Regulation of GSV translocation by Insulin”. Of course, signaling pathways involved are a part of insulin regulation.
In my opinion, everything about TUG also needs to be collected in one informative section.
Information on GLUT4 recycling is completely unnecessarily divided into several sections of the article, namely “Trafficking and Reconstitution of Irvs After Insulin Stimulation”, “Endocytosis: Recycling the Transporter”, and “Reconstitution of Gsvs After Insulin Stimulation”. Why? It would be logical to join them.
Similarly, sections “Involvement of Actin Cytoskeleton Remodelling In Gsv Trafficking” and “Insulin-Induced F-Actin Remodelling and Glut4 Vesicle Translocation” need to be joined.
- InFigure 2, in addition to GSV trafficking, you schematically depicted vasopressin inactivation. In my opinion, this is absolutely wrong. Vasopressin is an antidiuretic hormone that acts in the collecting ducts of the kidney, while GLUT4 functions in muscles and adipose tissue. There is no way they can functionally intersect in one cell, as shown in your scheme. Both listed insulin-driven phenomena exist in vivo, but not in one cell. It is advisable to delete vasopressin from that image.
- There are some “banalities” in the text of the review, which do not provide any significant specific information.
For instance: “Actin exists in two states: the monomeric globular form (G-actin) and the filamentous polymerized form (F-actin). F-actin remodelling involves continuous turnover, with polymerization at the barbed end and depolymerization at the pointed end. This dynamic is orchestrated by an array of regulatory proteins, including the Arp2/3 complex for nucleation, and severing proteins such as cofilin and gelsolin….. The key player in this process is the Arp2/3 complex, which consists of seven subunits (ARP2, ARP3, and ARPC1-5), which promotes the nucleation of a branched actin filament network”. This can be found in any cell biology textbook, does not help with the review subject and is very boring to read. I would suggest to remove these cytoskeleton elementary from your manuscript.
Another example: “Maintaining systemic blood glucose homeostasis after a meal requires rapid insulin secretion from pancreatic b-cells and a coordinated response in insulin-sensitive organs. Insulin sensitivity, the capacity to normalize blood glucose at a given insulin concentration, is a complex, system-wide property involving reduced hepatic glucose production, enhanced glucose uptake by muscle and adipose tissues, inhibition of adipose lipolysis, and promotion of lipid storage in the liver and fat deposits”. All of this is true, but it is absolutely not a cutting edge of research.
- Figure 3. In the caption to the picture, I read: “After internalization, GLUT4 can return to the plasma membrane via two main recycling routes. A rapid pathway through sorting endosomes, or a slower pathway through recycling endosomes”. With that, in the drawing I did not find any sorting endosome. There is an early endosome in the scheme, there is a recycling endosome, but there is no sorting endosome. Please fix that.
- The formatting of the manuscript, especially the formatting of the section headings and their numbering, is in complete confusion. Please correct this inaccuracy.
Author Response
Comments and Suggestions for Authors
The manuscript by Hana Drobiova and co-authors is a review analyzing the molecular mechanisms of insulin-dependent trafficking of glucose transporter GLUT4 in insulin-sensitive cells such as skeletal muscles and adipocytes. In these cells, the translocation of GLUT4 into the plasma membrane is carried out in a specialized type of vesicles, GLUT4 storage vesicles (GSVs). In the review, the authors consider in detail various aspects of GSV functioning, such as the biogenesis and molecular composition of GSVs, the mechanisms of their retention in the cytoplasm under basal conditions, and the signalling pathways of insulin that control GSV trafficking. The authors analyse in great detail the protein machinery mediating the sequential stages of GSVs exocytosis, including approach, tethering, docking, and fusion of GSVs with the plasma membrane. The review pays also much attention to the role of the cytoskeleton in GSV translocation. Next, the review examines the various currently proposed concepts of molecular mechanisms of GSV dysfunction in insulin resistance.
Obviously, the range of problems considered in the review is highly relevant for modern molecular medicine, given the high incidence of type 2 diabetes in the human population. In the text of the manuscript, the authors demonstrated a deep knowledge of the subject under consideration and a deep fundamental analysis of the review topic.
We thank the reviewer for his/her encouraging comment. We are pleased that the reviewer found our review to be comprehensive and relevant to the field of molecular medicine. The reviewer’s feedback is highly motivating and reinforces the value of our work.
Nevertheless, in my opinion, the manuscript is not free from a number of serious shortcomings that need to be addressed before it can be published. In fact, I think the whole paper should be re-written and re-structured. My comments for the authors are listed below.
- The quality of any review is largely determined by the choice of information selected for analysis. The translocation of GLUT4 and its disruption in insulin resistance is a hot topic on which hundreds of articles are published annually. This manuscript contains 282 references (quite a substantial number), of which 21 papers (7%) were published before 1995, 78 papers (28%) were published in the decade 1995-2004, 130 papers (46%) appeared between 2005 and 2014, and only 53 papers (19%) appeared in the last decade 2015-2025. In sum, 74% of publications analyzed in this manuscript were published 30 to 10 years ago!! In fact, this choice of information sources makes this manuscript belong more to the history of science than to current research. I understand that the basic concepts of the insulin-dependent GSV translocation were indeed mainly formulated about 20-25 years ago. But a review in a highly-rated journal should describe the current state of the art, rather than facts known for decades. In my opinion, there is no need to characterize long-known things in such detail, making hundreds of references to old and very old works. I highly recommend that the authors update the review, significantly shifting the proportion towards citation of research from the last decade.
We thank the reviewer for this constructive comment. We acknowledge that a contemporary review should prioritize the most recent developments in the field. However, our review was designed to serve as a comprehensive reference that not only integrates recent findings but also orients new researchers to the field by connecting classical work with emerging insights. Therefore, our review specifically focuses on discussing the composition, biogenesis, and intracellular trafficking of GSVs. The mechanistic basis of these processes was discovered during the late 1990s and early 2000s. These foundational studies provided the molecular architecture, that remain biologically valid and are frequently cited in current high-impact studies, forming the basis of ongoing research till now. Therefore, the significant portion of the cited literature that dates before 2015 reflects the historical development and enduring relevance of foundational discoveries in GLUT4 trafficking. Moreover, the fundamental principles of GLUT4 storage vesicle (GSV) dynamics have not changed drastically; rather, recent advances have refined these pathways in the context of metabolic diseases, new imaging tools, and therapeutic interventions.
Nonetheless, we have done a careful re-evaluation of the literature, as suggested, and have updated the manuscript to incorporate several more recent studies where relevant [Page 10, Lines 295-298; Page 21, Lines 622-629] thereby enriching the manuscript’s relevance to current and future investigations while maintaining historical depth and conceptual clarity.
- This manuscript is poorly structured and, as a result, very long. Many things are repeated twice in different sections. The division into chapters made in this version does not seem logical enough. I would suggest the following changes:
We agree with the reviewer, the MS may have some redundancy. Therefore, we followed the reviewer’s recommendations.
I think, section «Structural and Biochemical Characteristics of GSVs» can be easily merged with section “Biogenesis of Irvs” and section “Targeting of The Newly Synthesized Irv Proteins” since in all cases authors are talking about essentially the same processes and molecules, and resident protein targeting is a part of GSV biogenesis. It would be advisable to name this section "Biogenesis and molecular composition of GSVs”. The text of this section could be significantly shorter than the three sections in the current version. Also, it would be wise to add a table with information on the composition and known functions of the proteins contained in GSVs.
Similarly, sections “Regulation By Insulin” and “Insulin Pathway Involved In Gsv Translocation” also can be merged and shortened, collectively named as “Regulation of GSV translocation by Insulin”. Of course, signaling pathways involved are a part of insulin regulation.
In my opinion, everything about TUG also needs to be collected in one informative section.
Information on GLUT4 recycling is completely unnecessarily divided into several sections of the article, namely “Trafficking and Reconstitution of Irvs After Insulin Stimulation”, “Endocytosis: Recycling the Transporter”, and “Reconstitution of Gsvs After Insulin Stimulation”. Why? It would be logical to join them.
Similarly, sections “Involvement of Actin Cytoskeleton Remodelling In Gsv Trafficking” and “Insulin-Induced F-Actin Remodelling and Glut4 Vesicle Translocation” need to be joined.
We thank the reviewer for these insightful suggestions regarding the structure of the manuscript. We agree with the reviewer, the MS may have some redundancy. Therefore, we followed the reviewer’s recommendations to improve the manuscript organization which made it more readable. We have significantly restructured the manuscript. We have merged the suggested sections and have added a summary table outlining the composition and known functions of key GSV proteins, as recommended. Please see both the track-changes version of the manuscript with both all markup and simple markup version.
- In Figure 2, in addition to GSV trafficking, you schematically depicted vasopressin inactivation. In my opinion, this is absolutely wrong.
We thank the reviewer for highlighting this important physiological process. In order to keep the focus on the role of TUG cleavage in GSV translocation, we have eliminated the depiction of IRAP and LRP1 functions from the figure. At the same time, we would like to note that the concept initially illustrated in the figure is scientifically valid. As IRAP, within GSVs, translocate to the plasma membrane in response to insulin, it can cleave extracellular substrates, including vasopressin. In fact, vasopressin was the first physiological substrate identified as stated by the seminal study by Wallis et al.,{Wallis, 2007 #565}. We appreciate the reviewer’s careful attention and believe that this revision improves the clarity of the figure.
- There are some “banalities” in the text of the review, which do not provide any significant specific information.
For instance: “Actin exists in two states: the monomeric globular form (G-actin) and the filamentous polymerized form (F-actin). F-actin remodelling involves continuous turnover, with polymerization at the barbed end and depolymerization at the pointed end. This dynamic is orchestrated by an array of regulatory proteins, including the Arp2/3 complex for nucleation, and severing proteins such as cofilin and gelsolin. The key player in this process is the Arp2/3 complex, which consists of seven subunits (ARP2, ARP3, and ARPC1-5), which promotes the nucleation of a branched actin filament network”. This can be found in any cell biology textbook, does not help with the review subject and is very boring to read. I would suggest to remove these cytoskeleton elementary from your manuscript.
Another example: “Maintaining systemic blood glucose homeostasis after a meal requires rapid insulin secretion from pancreatic b-cells and a coordinated response in insulin-sensitive organs. Insulin sensitivity, the capacity to normalize blood glucose at a given insulin concentration, is a complex, system-wide property involving reduced hepatic glucose production, enhanced glucose uptake by muscle and adipose tissues, inhibition of adipose lipolysis, and promotion of lipid storage in the liver and fat deposits”. All of this is true, but it is absolutely not a cutting edge of research.
We thank the reviewer for this helpful observation. We have carefully revised the manuscript to remove or summarize the overly general background content. The text now focuses on the mechanistic aspects relevant to GSV trafficking.
- Figure 3. In the caption to the picture, I read: “After internalization, GLUT4 can return to the plasma membrane via two main recycling routes. A rapid pathway through sorting endosomes, or a slower pathway through recycling endosomes”. With that, in the drawing I did not find any sorting endosome. There is an early endosome in the scheme, there is a recycling endosome, but there is no sorting endosome. Please fix that.
We thank the reviewer for noting this discrepancy. We have corrected the figure, replacing the early endosome with the sorting endosome in the figure.
- The formatting of the manuscript, especially the formatting of the section headings and their numbering, is in complete confusion. Please correct this inaccuracy.
We thank the reviewer for this important comment. We have corrected the formatting of the headings and subheadings. We believe that the confusion is cleared and that this improves the clarity and readability of the review article.

Round 2
Reviewer 2 Report
Comments and Suggestions for Authors
Dear authors,
you have performed a great and thorough job of correcting the manuscript in accordance with the reviewers' recommendations. I thank you for this labour. As a result, the quality of your review, its consistency and scientific significance have increased significantly. I believe the article can be published in its current form.